# Effect of Spatiotemporal Parameters on the Gait of Children Aged from 6 to 12 Years in Podiatric Tests: A Cross Sectional Study

**DOI:** 10.3390/healthcare11050708

**Published:** 2023-02-27

**Authors:** Magdalena Martinez-Rico, Ana Belen Ortega-Avila, Consolacion Pineda-Galan, Gabriel Gijon-Nogueron, Manuel Pardo Rios, Raquel Alabua-Dasi, Ana Marchena-Rodriguez

**Affiliations:** 1Department of Nursing and Podiatry, University of Malaga, 29016 Malaga, Spain; 2KU Leuven, Department of Rehabilitation Sciences, Musculoskeletal Rehabilitation Research Group, Campus Brugge, Spoorwegstraat 12, 8200 Bruges, Belgium; 3Instituto deInvestigacion Biomedicina de Malaga (IBIMA), 29590 Malaga, Spain; 4Department of Phisiotherapy, University of Malaga, 29016 Malaga, Spain; 5Department of Podiatry, Faculty of Health Sciences, Campus de Los Jerónimos, Guadalupe, Universidad Católica San An tonio de Murcia, 30107 Murcia, Spain

**Keywords:** children, gait, foot, outcomes

## Abstract

The use of lower limb tests in the paediatric population is of great importance for diagnostic evaluations. The aim of this study is to understand the relationship between the tests performed on the feet and ankles, covering all of its planes, and the spatiotemporal parameters of children’s gait. Methods: It is a cross-sectional observational study. Children aged between 6 and 12 years participated. Measurements were carried out in 2022. An analysis of three tests used to assess the feet and ankles (FPI, the ankle lunge test, and the lunge test), as well as a kinematic analysis of gait using OptoGait as a measurement tool, was performed. Results: The spatiotemporal parameters show how Jack’s Test is significant in the propulsion phase in its % parameter, with a *p*-value of 0.05 and a mean difference of 0.67%. Additionally, in the lunge test, we studied the % of midstance in the left foot, with a mean difference between the positive test and the 10 cm test of 10.76 (*p* value of 0.04). Conclusions: The diagnostic analysis of the functional limitation of the first toe (Jack’s test) is correlated with the spaciotemporal parameter of propulsion, as well as the lunge test, which is also correlated with the midstance phase of gait.

## 1. Introduction

The use of lower limb tests in the paediatric population is of great importance for diagnostic evaluation and treatment. The use of these tests to assess foot and ankle functions is a controversial topic, showing a lack of consensus on how foot and ankle functions should be measured, defined, or assessed [1].

Therefore, high-reliability morpho-functional tests of the foot and ankle should be used [2,3] in order to establish an association between the morpho-functional variants [4] and other variables, such as weight [5], laxity [6], and physical activity [7] or gait.

A child’s gait can be influenced by many intrinsic factors: limb length, joint range, muscle tone [8], neuromuscular diseases [9], and also, by extrinsic factors: footwear, clothing, or carrying loads, which may change their walking pattern [10]. Measures of spatio-temporal gait parameters are used to identify and diagnose walking difficulties and may also determine the prognosis [11].

Although in many cases we are not able to interrelate the different existing diagnostic tests of the foot, this is of vital importance to facilitate subsequent treatment, so determining this interrelation should be a priority issue in lower extremity examinations.

Despite this, fundamental information on joint and foot mechanics, as written by K, Deschamps et al., on typically developing children and children with pathological condition has not yet been provided. Indeed, the joint kinetic profiles that have been reported in the past focus on the major joints of the lower extremity (e.g., the hip, the knee, and the ankle) [12] Therefore, joints distal to the ankle play a key role in energy absorption and generation during gait [13,14]. Technical limitation is the main reason why this biomechanical perspective remains understudied [15,16].

The OptoGait system is more readily accessed and can be used in primary care consultation. It is based on a photoelectric cell and is validated for the assessment of the phases of gait in clinical and research settings [17,18]. The coefficient of variation in method error values was low, ranging from 1.66% to 4.06%, and all the parameters presented standard errors of measurement between 2.17 and 5.96%, indicating a strong reliability.

OptoGait has demonstrated excellent reliability for the variables of step rate, step length, and contact time during treadmill and ground walking, as well as good test–retest reliability in healthy and injured adults [19]. However, very few studies have evaluated the spatiotemporal parameters of gait with this system in children [20,21].

As described above, there is a lack of information on the relationship between foot tests and gait parameters in the pediatric population to guide our clinical practice. The hypothesis was that foot tests could be related to gait analysis. Therefore, the aim of this study is to relate diagnostic tests on the foot, covering all of its planes, with the spatiotemporal parameters of children’s gait.

## 2. Method

### 2.1. Ethical Approval

The parent and/or legal guardian were provided with information about the study, and a statement attesting to informed consent was signed. The children were fully informed of the procedures involved and gave their consent. All the procedures were in accordance with the ethical standards of the institution, and the experimental protocol was approved by a named institutional of the University of Malaga (CEUMA 91/2016H) and with the 1964 Helsinki declaration.

### 2.2. Study Design

It is a cross-sectional observational study, in which the Strengthening Reporting of Observational Studies in Epidemiology (STROBE) criteria were followed.

#### Sample Size Calculation

To calculate the sample size, we used Epidat 4.2 software (Epidemiology Service of the General Directorate of Public Health of the Consellería de Sanidade (Xunta de Galicia)), and we used the paper by Montes-Alguacil et al. [22] to obtain mean and standard deviations for the main outcome’s variables: the heel contact phase, the flat foot phase, and the propulsion phase. The study was designed to detect changes exceeding 0.8 (high effect size) with a type I error of 0.05 and a type II error of 0.2. This calculation used a necessary sample size of 48 subjects, although, in fact, 50 were recruited to cover any potential missing data.

### 2.3. Participants

Children aged between 6 and 12 years participated. Measurements were carried out in 2022. The participants were recruited at the Faculty of Health Science from the University of Malaga (Spain).

The inclusion criteria were participants aged between 6 and 12 years old and those not experiencing any foot pain at the time of the assessment. Participants who had any of the following conditions were excluded from the study: recent damage of the lower limbs, congenital structural alterations that affect distal areas of the ankle joint, as well as those cases with pathological flat feet caused by cerebral palsy, surgical treatments in the foot or lower extremities, or affectations of a genetic, neurological, or muscular nature.

### 2.4. Procedure

The test was performed by two different clinicians. Both were blinded to themselves and to each other’s results. One performed the 3 tests used to evaluate the foot and ankle (FPI, the ankle lunge test, and Jack’s test), and the other one performed the gait analysis (MMR) using the OptoGait gait cycle measurement tool.

#### 2.4.1. Foot Posture Index (FPI)

The assessment of the foot posture was carried out by measuring the FPI with barefoot subjects in a relaxed standing position to facilitate the visual and physical inspection. The inter-examiner reliability for the FPI in the paediatric population reached a consistent weighted Kappa value (Kw = 0.86) in a sample of children aged between 5 and 16 years, and the categorization was performed using the criteria by Gijon-Nogueron et al. [23,24,25]

#### 2.4.2. Ankle Lunge Test

The range of ankle dorsiflexion was determined by the lunge test, which is a weight-bearing test of the range of ankle dorsiflexion when the knee is flexed. The participant was stood in a relaxed standing position on a solid, horizontal surface facing a vertical wall. The test foot was parallel with a tape measure secured to the floor with the second toe, centre of the heel, and knee perpendicular to a wall. To promote upright balance during the test, the opposite limb was positioned approximately 1 foot behind the test foot in a comfortable tandem stance, and the subjects placed their hands on the wall. The test involved the participant pushing their knee as far forward over the foot as possible, while keeping the heel on the ground. The maximum angle of advancement of the tibia relative to the vertical was recorded as a measure of ankle dorsiflexion using a digital inclinometer (Smart Tool™) applied to the anterior surface of the tibia. The intra-examiner intraclass correlation coefficients were 0.98, and the inter-examiner reliability reached an excellent value of weighted Kappa (Kw = 0.97) [26].

#### 2.4.3. Jack´s Test

In 1953, Jack described one of the first methods to evaluate the first MPJ in a weight-bearing position, and this method is still commonly used today. It is also known as the Hubscher manoeuvre [27].

The utility of the test assumes that restriction during the static manoeuvre is predictive of the functional limitation of this joint during gait. The test involves the examiner manually dorsiflexing the hallux, while the patient stands in a relaxed double-limb stance position. 

A normal response is for these structures to ‘tighten’ and the medial longitudinal arch to rise. This response is commonly referred to as the ‘windlass mechanism’. A failed test was recorded when the examiner was unable to dorsiflex the hallux from the weightbearing surface without the application of excessive force. 

The utility of the test assumes that restriction during the static manoeuvre is predictive of functional limitation of this joint during gait. The intra-examiner intraclass correlation coefficients were 0.89.

#### 2.4.4. Spaciotemporal Gait by OptoGait System

The gait parameters were assessed using the OptoGait^®^ portable photocell system [17,18]. This system provides real-time numerical parameters related to stepping, running, and jumping.

Previously, the participants were instructed to walk barefoot, as per there normal walking behavior, for a distance of five meters between two parallel bars. Six to eight strides are sufficient to obtain representative data for unimpaired adults [17], and in our case, ten strides were measured. After three trials, the data were acquired. A highly experienced podiatrist (MMR) (with more than 1000 Optogait^®^ tests examinations performed) controlled the measurement process at all times.

The software used was OptoGait v.1.11.1.0. The Optogait system was calibrated and checked for accuracy at all times to provide an exhaustive and reliable measurement of the spatiotemporal phases of the gait cycle. Considering the heel contact phase (Phase 1), this is the time from the initial ground contact (1 LED activated is needed to be considered) to the foot flattening (the number of LEDs activated stays steady ±2 LEDs). Footflat phase (Phase 2) is the time from foot flattening to the initial take-off, and the propulsive phase (Phase 3) is the initial take-off until the end of the motion.

### 2.5. Statistical Analysis

The descriptive statistics obtained included measures of central tendency and dispersion and the distribution of percentages.

An exploratory analysis including the Kolmogorov–Smirnov test and by examining symmetry and kurtosis was performed to confirm the normality of the distributions. Subsequently, a bivariate analysis of the differences of the means using Student’s t test was applied to evaluate the differences in the gait parameters according to the results of Jack’s test.

The differences in the gait parameters were identified by ANOVA according to the four FPI and the three lunge test groups established. The homoscedasticity of the distributions was determined by the Levene test. In addition, the Browne–Forsythe test of robustness was applied, and a post hoc analysis performed using the Bonferroni test. The level of statistical significance was 95% in all the cases, and all the analyses were conducted using SPSS v.23 statistical software (SPSS Inc., Chicago, IL, USA).

## 3. Results

The sample in this study was composed of 50 healthy school children, with 29 (58%) girls and 21 (42%) boys aged 6–12 years (mean age: 8.96 years, SD: 1.83). The mean BMIs were 18.45 kg/m^2^ (SD:3.30) for the girls and 18.74 kg/m^2^ (SD:3.52) for the boys. The difference between the genders was not statistically significant (t = −0.23; *p* = 0.814).

According to the results observed the Jack´s test showed, for the left foot, 42 negative results for 8 positive results, and for the right foot, 44 negative and 6 positive results. Regarding the lunge test, for the left foot, 12 results were positive (unable to performed the test) and 38 results were negative (20 negative, 5 cm, and 18 negative, 10 cm), and for the right foot, 10 results were positive and 40 results were negative (22 negative, 5 cm, and 18 negative, 10 cm) (Table 1).

The spaciotemporal parameters show how the results of Jack’s test are significant in the propulsion phase in its % parameter, with a *p*-value of 0.05 and a mean difference of 0,67% (Table 2).

If we divide the groups according to age, significant differences are only observed for the same measures as those in the global analysis (*p* = 0.038 left propulsive phase (%)), with the rest having *p*-values greater than 0.05 (Figure 1).

Regarding the lunge’s tests (Table 3), according to age, significant differences were only observed in the 10-year-old sample in the flat foot phase on the right foot (*p* = 0.022, with the rest of them having *p*-values greater than 0.05 (Figure 2).

Finally, a significance is observed in the relationship between the gait parameters and the foot posture, following the foot posture index. We found a significance in the step length between the pronated posture and normal posture in the right foot (with a mean difference of 5.51 cm and a *p*-value of 0.05), as well as in the contact phase (with a mean difference of 0.63 sec and a *p*-value of 0.05) (Table 4).

Regarding the FPI test, according to age, significant differences were only observed in the 8-year-old sample in the flat foot phase on the right foot (*p* = 0.009, the rest of them having *p*-values greater than 0.05 (Figure 3).

## 4. Discussion

The tests used for the evaluation of the foot can be used to identify a relationship between them and the different phases of the gait cycle, verifying which of them they influence. The objective of this study was to relate the tests performed on the foot and covering all its planes with the spatiotemporal parameters of children’s gait.

The gait cycle begins when the foot makes contact with the ground, and it ends when the same foot makes contact with the ground again. The first ray is fundamental both in the full stance phase, in which it will serve as a mobile adapter on the irregularities of the ground, forming an internal longitudinal arch, and in the propulsive phase, whose function will be to become a rigid segment capable of transferring the weight of the body forward [28].

The windlass mechanism [13] is closely related to the propulsion phase, since the correct movement of the first metatarsophalangeal joint, together with the locking of the midtarsal joint makes this gait phase functional [29].

This windlass mechanism creates tension in the plantar aponeurosis, with tensile forces approaching 100% of the body weight.

Although it is highly variable, the arch rises rapidly during the late stance phase, and the navicular demonstrates an average rise of 6 mm during late push-off. Depending on the foot model used for the gait analysis, dorsiflexion of the first metatarsophalangeal joint averages around 30–50° during this same period [30].

Jack’s test is a test that has been devised to clinically evaluate the function of this first ray, therefore, a positive test would reflect an inability to dorsiflex this first metatarsophalangeal joint (causing what is known as functional hallux limitus), and therefore, an alteration of this windlass mechanism mentioned above [31].

This is observed in our results, which show an increase in the percentage of the propulsion phase in patients with a positive Jack´s test (difference 0.67 and a *p*-value of 0.05). Those patients who were unable to perform Jack’s test properly, and in whom, therefore, their windlass mechanism does not work properly, presented a higher percentage in the propulsion phase. Although this is only in the left foot, which leads us to believe that in many cases there are also differences between the stance phases of both feet, which would suggest a more exhaustive, unilateral analysis of gait [14].

However, this contrasts with the opinion of some authors who defend that the presence of limitation of dorsal flexion in this test is not indicative of the limitation of this movement in gait, but that there is a relationship between the pronation of the foot and the limitation of the first metatarsophalangeal joint, which could explain another part of our results [31,32]. However, the study population were patients older than 18 years, and they focused only on the movement of the first metatarsophalangeal joint and not on the entire windlass mechanism. These may be variables that have influenced the conclusion.

Another very important structure in the human gait cycle is the Achilles-calcaneal-plantar system. A weight-bearing motion in the first MP joint depends on structures that are not located at the joint itself, but more proximal ones. Among these structures, the Achilles-calcaneal-plantar system and the medial column of the foot are mainly responsible for optimally setting the first MPJ to provide for anteromedial support to the foot during the third rocker or propulsive phase of gait; this requires adequate passive dorsiflexion of the joint, while the hallux is purchasing the ground, and the verticalized first metatarsal is axially loading the hallux sesamoid complex [33].

At this point, the function of the first ray is very important. The first 20° of dorsiflexion are performed thanks to the triceps surae, which raises the heel and makes the first metatarsal initiate this plantar flexion movement. Limited passive dorsiflexion of the first MP joint limits the motion in the sagittal plane, which is necessary for the forward progression of the body during gait [34].

During the second rocker, the tibia must glide forward on the ankle to allow the body’s center of mass to progress from an initial position posterior to the supporting foot to a final position that is anterior to it. A restriction of ankle passive dorsiflexion during the second rocker will increase dorsiflexion moments at the forefoot. Under normal conditions, during the second rocker, the position of the foot must change from pronation to supination; from a relaxed and cushioning conformation to a rigid and propulsive one. If the ankle is unable to provide the necessary passive dorsiflexion for the centre of mass to be placed in front of its vertical plane, one of the ways to achieve these degrees of dorsiflexion is the pronation of the foot [33].

Therefore, our results reinforce all this, showing a relationship between the lunge test (which, as mentioned above, is predictive of dorsiflexion limitation) and an alteration in the full foot contact phases and in the propulsion phase.

Regarding these two phases of gait, we observed a decrease in the time of the total contact phase, decreasing the time that the foot is on the floor in those patients tested at 5 cm, as well as an increase in the time of the propulsion phase in those patients tested at 10 cm.

The results of the lunge test and Jack’s test could be a starting point to obtain information from the existing literature, which describe that: an increase in the dorsiflexion of the ankle during the end of stance phase produces stress on the Achilles tendon and a decrease in plantarflexion during the propulsion period [35,36].

Finally, another of the tests analysed, which also had an impact on gait, was the analysis of foot posture during pronation or supination according to the FPI, where it is observed that the mid-contact phase influences the result, producing a longer contact time in pronated feet than it does in neutral feet by 0.06 s. These data, although not obtained using the same measuring tool, are similar to those proposed by Ryan Mahaffey et al. [37], who proposed the increase in the pronation phase of the midfoot. Caravaggi et al. [38] showed significant postural and kinematic alterations in the midtarsal and tarso-metarsal joints of adolescents with planus valgus feet. The objective identification and quantification of planus valgus foot alterations via non-invasive gait-analysis is relevant to improving the diagnosis of this condition and to evaluating the effect of conservative treatments and of surgical corrections by different techniques.

Therefore, both our results and those of other authors show that a pronated foot (measured using the FPI test), as well as a limitation of the dorsal flexion of the ankle (measured using the lunge test) and a limitation of the movement of the first metatarsophalangeal joint (measured using Jack’s test) are factors that influence different moments in the gait cycle. In our case, in addition, it is demonstrated that the tests used in the foot are predictive of these presentations of gait and that, therefore, a relationship can be established between these tests and the spatiotemporal parameters of gait, and it should be studied in a different population such as the child population.

The clinical implication of this study is related to the exploration of a new option for assessing feet and the gait because clinicians will not always have the tools to evaluate the gait parameters. Therefore, these tests could serve as a proxy of this measure, but they will never be a replacement for it.

All of these results should be approached with caution, since they have limitations. The main one is the size of the sample, since it is a convenience sample. This sample was obtained in an exploratory manner, and we had to classify the participants in different subgroups of age, sex, and parameters such as physical activity and weight. In addition, it has the limitation of being a cross-sectional study that always provides punctual data, and not an evolution of the data over time, which is appropriate. Even so, to our knowledge, it is the first study that has begun to relate spatiotemporal parameters with foot tests in children in order to answer one of the great questions of clinicians, which is the interrelation between gait and diagnostic tests.

## 5. Conclusions

The diagnostic analysis tests of the functional limitation of the first toe (Jack’s test) is correlated with the spatiotemporal parameter of propulsion, as well as the lunge test, which also correlates with the midstance phase of the gait, which in turn correlates with the posture of the foot, where an increase is observed in the contact time of the pronated feet.

## Figures and Tables

**Figure 1 healthcare-11-00708-f001:**
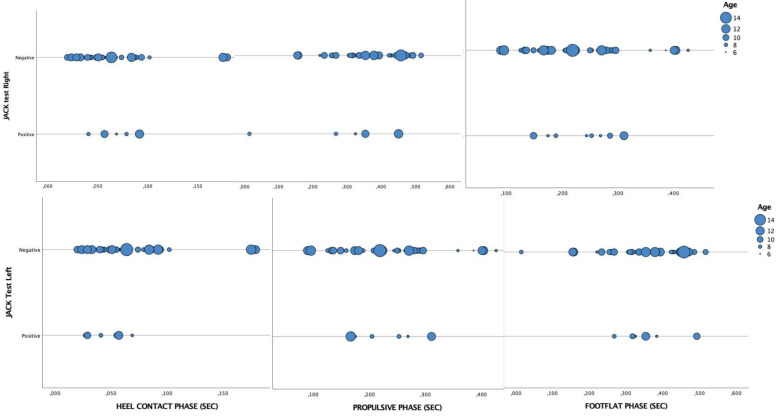
Scatter plot analysis of the gait parameters according to the results of Jack’s test groups and age.

**Figure 2 healthcare-11-00708-f002:**
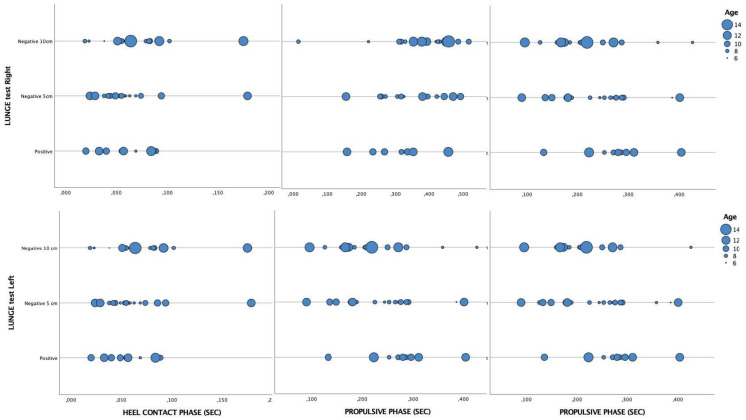
Scatter plot analysis of the gait parameters according to the results lunge test groups and age.

**Figure 3 healthcare-11-00708-f003:**
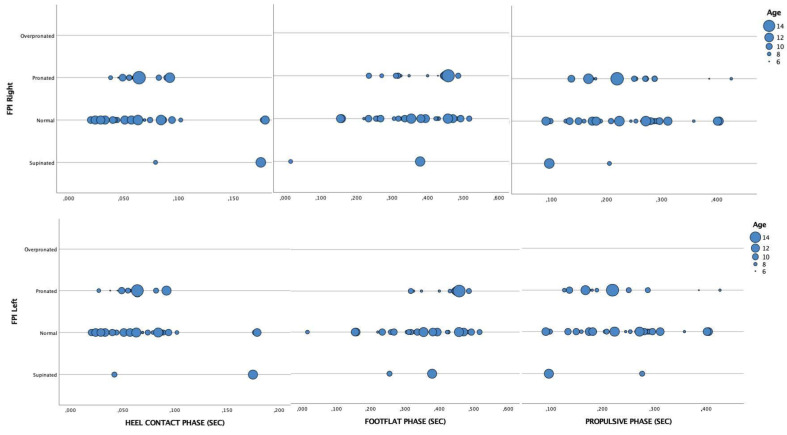
Scatter plot analysis of the gait parameters according to the results from the FPI test groups and age.

**Table 1 healthcare-11-00708-t001:** Characteristics of the sample tests.

		MALE		FEMALE	
		Frecuency (%)			
		Left	Right	Left	Right
**LUNGE’S TEST**	Positive	6 (28.6)	5 (23.8)	6 (20.7)	5 (17.2)
	Negative 5 cm	6 (28.6)	8 (38.1)	14 (48.3)	14 (48.3)
	Negative 10 cm	9 (42.9)	8 (38.1)	9 (31)	10 (34.5)
**JACK’S TEST**	Positive	4 (19)	1 (4.8)	4 (13.8)	5 (17.2)
	Negative	17 (81)	20 (95.2)	25 (86.2)	24 (82.8)
**FPI CATEGORIZATION**	Supinated	0	0	1 (3.4)	2 (6.9)
	Normal	16 (76.2)	15 (71.4)	21 (72.4)	19 (65.5)
	Pronated	5 (23.8)	6 (28.6)	7 (24.1)	8 (27.6)

**Table 2 healthcare-11-00708-t002:** Student’s t test analysis of the gait parameters according to the results of Jack´s test (Cm: centimetres; Seg: second; N: numbers; SD: standard deviation).

	JACK RIGHT						JACK LEFT				
		N	Mean	SD	Dif. Mean	*p* Value	N	Mean	SD	Dif. Mean	*p* Value
**STEP LENGTH (CM)**	Positive	6	43.22	9.93	0.10	0.577	8	46.04	10.11	3.46	0.417
	Negative	44	43.12	8.48			42	42.57	8.25		
**HEEL CONTACT PHASE (SEC)**	Positive	6	0.07	0.02	0.00	0.292	8	0.05	0.02	−0.01	0.173
	Negative	44	0.06	0.04			42	0.06	0.04		
**HEEL CONTACT PHASE (%)**	Positive	6	9.90	2.11	0.13	0.165	8	8.18	2.35	−1.92	0.179
	Negative	44	9.77	6.02			42	10.09	6.08		
**FOOTFLAT PHASE (SEC)**	Positive	6	0.29	0.15	−0.06	0.36	8	0.35	0.07	0.00	0.128
	Negative	44	0.35	0.09			42	0.34	0.11		
**FOOTFLAT PHASE (%)**	Positive	6	55.77	7.53	1.33	0.333	8	55.08	8.37	0.57	0.185
	Negative	44	54.44	11.10			42	54.51	11.16		
**PROPULSIVE PHASE (SEC)**	Positive	6	0.23	0.06	0.00	0.357	8	0.23	0.06	0.00	0.279
	Negative	44	0.23	0.08			42	0.23	0.08		
**PROPULSIVE PHASE (%)**	Positive	6	34.33	8.89	−1.92	0.328	8	36.59	7.57	**0.67 ***	**0.05**
	Negative	44	36.25	12.89			42	35.91	13.21		

* statistically significant test result (*p* ≤ 0.05).

**Table 3 healthcare-11-00708-t003:** ANOVA test analysis of the gait parameters according to the results of the lunge test groups (Cm: centimetres; Seg: second; N: numbers; SD: standard deviation; CI: Confidence Interval).

			LUNGE LEFT				LUNGE RIGHT			
			Dif. Mean	*p* Value	C.I. 95%		Dif. Mean	*p* Value	C.I. 95%	
					Lower	Higher			Lower	higher
**STEP LENGTH (CM)**	Positive	Negative 5 cm	2.89	0.74	−7.24	13.02	3.17	0.69	−6.64	12.98
		Negative 10 cm	0.64	0.99	−9.77	11.05	2.43	0.81	−7.54	12.40
	Negative 5 cm	Negative 10 cm	−2.25	0.64	−8.32	3.82	−0.74	0.95	−6.86	5.38
**HEEL CONTACT PHASE (SEC)**	Positive	Negative 5 cm	−0.01	0.77	−0.03	0.02	0.00	0.98	−0.03	0.02
		Negative 10 cm	−0.02	0.16	−0.06	0.01	−0.02	0.30	−0.05	0.01
	Negative 5 cm	Negative 10 cm	−0.02	0.38	−0.05	0.01	−0.02	0.40	−0.05	0.01
**HEEL CONTACT PHASE (%)**	Positive	Negative 5 cm	−0.88	0.88	−5.40	3.64	0.58	0.96	−4.60	5.77
		Negative 10 cm	−2.64	0.42	−7.80	2.52	−1.34	0.81	−6.78	4.10
	Negative 5 cm	Negative 10 cm	−1.76	0.65	−6.59	3.08	−1.93	0.56	−6.48	2.63
**FOOTFLAT PHASE (SEC)**	Positive	Negative 5 cm	−0.04	0.49	−0.13	0.05	−0.07	0.09	−0.15	0.01
		Negative 10 cm	−0.07	0.18	−0.17	0.03	−0.09	0.07	−0.18	0.00
	Negative 5 cm	Negative 10 cm	−0.03	0.59	−0.11	0.05	−0.02	0.88	−0.09	0.06
**FOOTFLAT PHASE**	Positive	Negative 5 cm	−6.81	0.27	−17.66	4.03	−9.69 *	0.04	−18.91	−0.49
		Negative 10 cm	−10.76 *	0.04	−20.99	−0.55	−11.86 *	0.01	−20.56	−3.17
	Negative 5 cm	Negative 10 cm	−3.96	0.38	−11.14	3.23	−2.17	0.78	−9.98	5.64
**PROPULSIVE PHASE (SEC)**	Positive	Negative 5 cm	0.05	0.19	−0.02	0.11	0.05	0.18	−0.02	0.12
		Negative 10 cm	0.06	0.08	−0.01	0.13	0.06	0.11	−0.01	0.13
	Negative 5 cm	Negative 10 cm	0.01	0.88	−0.05	0.08	0.01	0.95	−0.06	0.07
**PROPULSIVE PHASE (%)**	Positive	Negative 5 cm	8.81	0.18	−3.24	20.86	10.25	0.08	−0.98	21.48
		Negative 10 cm	**13.88 ***	**0.02**	2.33	25.44	**13.74 ***	**0.01**	2.82	24.67
	Negative 5 cm	Negative 10 cm	5.07	0.31	−3.32	13.47	3.50	0.61	−5.41	12.40

* Statistically significant test result (*p* ≤ 0.05).

**Table 4 healthcare-11-00708-t004:** ANOVA test analysis of the gait parameters according to the results FPI test groups (Cm: centimetres; Seg: second; N: numbers; SD: standard deviation; CI: Confidence Interval).

			FPI RIGHT				FPI LEFT			
			Dif. Mean	*p* Value	C.I. 95%		Dif. Mean	*p* Value	C.I. 95%	
					Lower	Higher			Lower	Higher
**STEP LENGTH (CM)**	Supinated	Normal	−3.30	0.80	−59.32	52.71	0.04	1.00	−123.68	123.76
		Pronated	2.21	0.90	−46.74	51.15	5.81	0.78	−100.32	111.94
	Normal	Pronated	**5.51 ***	**0.05**	−0.36	11.38	5.77	0.08	−0.63	12.17
**HEEL CONTACT PHASE (SEC)**	Supinated	Normal	0.07	0.56	−0.78	0.91	0.05	0.80	−1.17	1.27
		Pronated	0.07	0.57	−0.81	0.95	0.05	0.78	−1.18	1.29
	Normal	Pronated	0.00	0.99	−0.02	0.02	0.00	0.85	−0.02	0.02
**HEEL CONTACT PHASE (%)**	Supinated	Normal	9.29	0.69	−159.64	178.22	7.93	0.78	−181.11	196.96
		Pronated	9.28	0.69	−159.96	178.51	9.20	0.73	−183.52	201.92
	Normal	Pronated	−0.01	1.00	−3.35	3.33	1.28	0.53	−1.58	4.13
**FOOTFLAT PHASE** **(SEC)**	Supinated	Normal	−0.15	0.76	−3.50	3.20	−0.01	0.98	−0.87	0.84
		Pronated	−0.17	0.72	−3.45	3.11	−0.08	0.61	−0.92	0.77
	Normal	Pronated	−0.02	0.74	−0.09	0.05	**−0.063 ***	**0.05**	−0.13	0.00
**FOOTFLAT PHASE** **(%)**	Supinated	Normal	7.03	0.32	−13.38	27.45	−1.02	0.99	−78.70	76.66
		Pronated	5.79	0.42	−12.44	24.02	−7.54	0.60	−76.60	61.52
	Normal	Pronated	−1.24	0.91	−8.76	6.28	−6.52	0.08	−13.75	0.71
**PROPULSIVE PHASE** **(SEC)**	Supinated	Normal	−0.10	0.45	−0.69	0.49	−0.04	0.91	−1.24	1.16
		Pronated	0.08	0.54	−0.74	0.90	0.05	0.86	−1.53	1.63
	Normal	Pronated	−0.02	0.79	−0.08	0.05	0.01	0.92	−0.07	0.09
**PROPULSIVE PHASE** **(%)**	Supinated	Normal	16.59	0.31	−58.42	91.61	7.73	0.89	−275.20	290.65
		Pronated	0.74	0.97	−7.52	9.01	6.03	0.17	−1.95	14.01
	Normal	Pronated	15.85	0.32	−54.41	86.11	1.70	0.99	−276.01	279.41

* Statistically significant test result (*p* ≤ 0.05).

## Data Availability

Not applicable.

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
