# Peer review of "Effect of Spatiotemporal Parameters on the Gait of Children Aged from 6 to 12 Years in Podiatric Tests: A Cross Sectional Study"

_healthcare, 2023, doi:10.3390/healthcare11050708_

Round 1

Reviewer 1 Report

The purpose of the study was to investigate: Relationship of spatiotemporal parameters on the gait of children aged 6 to 12 years with podiatric tests: A Cross-sectional study. The purpose of this cross-sectional study was to relate diagnostic tests on the foot covering all its planes with the spatiotemporal parameters of children's gait. The sample in this study was composed of 50 healthy schoolchildren between 6 and 12 years old. The participants performed the 3 tests used to evaluate the foot and 91 ankles (FPI, Ankle lunge test, and Jack’s test) and the gait analysis using the OptoGait gait cycle measurement tool.

We have considered the key question in this study: there are lots of anatomic variations, including foot length and width, between 6 and 12 years old. Children’s feet are rapidly growing in this period. We do not recommend analyzing the parameters average. Divided into small groups is suggested to reduce the variations. 

Author Response

Q. The purpose of the study was to investigate: Relationship of spatiotemporal parameters on the gait of children aged 6 to 12 years with podiatric tests: A Cross-sectional study. The purpose of this cross-sectional study was to relate diagnostic tests on the foot covering all its planes with the spatiotemporal parameters of children's gait. The sample in this study was composed of 50 healthy schoolchildren between 6 and 12 years old. The participants performed the 3 tests used to evaluate the foot and 91 ankles (FPI, Ankle lunge test, and Jack’s test) and the gait analysis using the OptoGait gait cycle measurement tool.

We have considered the key question in this study: there are lots of anatomic variations, including foot length and width, between 6 and 12 years old. Children’s feet are rapidly growing in this period. We do not recommend analyzing the parameters average. Divided into small groups is suggested to reduce the variations. 

A. Dear reviewer, thank you for your feedback, we have included new figures by age and new paragraphs of the results. 

Reviewer 2 Report

Thank you for study.   

The number of subjects in the article is quite small, but I think it will be not useful to evaluate Investigation of relationship of spatiotemporal parameters on the gait of chil-2 dren aged 6 to 12 years with podiatric tests.

In the introduction, What is the hypothesis of the study?

Power analysis should be done in the study,

In the 6-12 age group, the results without dividing them into subgroups may vary, and the foot structure can vary greatly in these age groups, along with bone and muscle development. The study should be divided into subgroups.

According to this study, it may not be correct to comment on the degree of dorsiflexion during walking, because there is no dorsiflexion degree taken and the gait was not evaluated in 3 dimensions, this part should be corrected in the discussion part.

Author Response

Q. The number of subjects in the article is quite small, but I think it will be not useful to evaluate Investigation of relationship of spatiotemporal parameters on the gait of children aged 6 to 12 years with podiatric tests.

In the introduction, What is the hypothesis of the study?

A. We have included new paragraphs with a hypothesis even if in the observational studies is better to use the objectives because we don’t have an action the participants

Q. Power analysis should be done in the study,

A. We have included a new paragraph about it

Q. In the 6-12 age group, the results without dividing them into subgroups may vary, and the foot structure can vary greatly in these age groups, along with bone and muscle development. The study should be divided into subgroups.

A. We have included new figures by age and new paragraphs of the results. 

Q. According to this study, it may not be correct to comment on the degree of dorsiflexion during walking, because there is no dorsiflexion degree taken and the gait was not evaluated in 3 dimensions, this part should be corrected in the discussion part.

A. We have changed this sentence

Reviewer 3 Report

Dear authors:
It has been a pleasure to review your paper “Relationship of spatiotemporal parameters on the gait of children aged 6 to 12 years with podiatric tests: A Cross-sectional study.”
This paper focuses on the analysis of resolving the relationship between the foot tests and the spatiotemporal gait analysis, it's important in this deficient area to determine if these tests, which the clinician use in their assessment, are related to the gait analysis because the treatment(insoles or strength exercises) will be with this results.
This paper includes new and important information about the relevance of the foot test as Jack's test, without comparison between the functional action and the gait analysis, noting that this test is correct for evaluating first-ray movement
The methodology is the section that is necessary to improve, with the below recommendations
method:
- Please can you improve the paragraph of gait analysis by optogait? I don't see clearly, what measure used.
- Can you clarify how do you determine of sample size?
And I can see that the discussion should be improved with the below recommendation.
Discussion:
- Can you improve the clinical implication related to health care? It's not clear.
The conclusions are related to the aim so it's important to continue with this line of research.
The tables show the information that the authors write in the rest of the paper and use in the discussion section.

Author Response

Q.  Please can you improve the paragraph on gait analysis by optogait? I don't see clearly, what measure was used.

A. We have included a new text

Q. - Can you clarify how you determine sample size?

A. We have included a new section

Q. And I can see that the discussion should be improved with the below recommendation.
Discussion:
- Can you improve the clinical implication related to health care? It's not clear.

A. We have included a new paragraph

Round 2

Reviewer 1 Report

Thank you for your responses. The writers have changed and revised the related issues in this article.

Reviewer 2 Report

Thank you for the necessary additions.